# Returning to Work after Breast Cancer: A One-Year Mixed-Methods Study

**DOI:** 10.3390/ijerph21081057

**Published:** 2024-08-13

**Authors:** Nicola Magnavita, Igor Meraglia, Daniela Andreina Terribile

**Affiliations:** 1Post-Graduate School of Occupational Health, Department of Life Sciences and Public Health, Università Cattolica del Sacro Cuore, 00168 Rome, Italy; igor.meraglia01@icatt.it; 2Multidisciplinary Breast Center, Dipartimento Scienze della Salute della Donna e del Bambino e di Sanità Pubblica, Fondazione Policlinico Universitario Agostino Gemelli IRCCS, Largo Agostino Gemelli 8, 00168 Rome, Italy; daniela.terribile@unicatt.it

**Keywords:** disability management, longitudinal study, welfare, sleep, anxiety, depression, fatigue, work organization, barriers, facilitators, workplace

## Abstract

Breast cancer (BC) is the most common invasive neoplasm and affects many women of working age. The return to work (RTW) of female survivors (BCSs) is associated with a better quality of life and longer survival. A tailored intervention to promote RTW was launched in 2022. A year later, the women were contacted to find out if RTW had occurred regularly and what their health conditions were compared to the baseline. BCSs reported excessive fatigue, poor sleep quality, anxiety, depression and reduced work ability; these parameters had not improved significantly compared to the baseline. Thematic analysis of the interviews confirmed the presence of personal, company, and societal factors that could hinder or favor RTW. The interviews demonstrated that, even in an economically developed country that has provided numerous benefits for BCSs, protection is not always effective. Personalized intervention seems necessary to complete the process of reintegrating BCSs into their future working careers.

## 1. Introduction

Breast cancer (BC) is the most frequent invasive neoplasm in females, with a global incidence of more than 2.3 million cases per year [1]. In the decade 2010–2019, the incidence of newly diagnosed cases increased by 0.5 percent annually, while there was a slight decrease in mortality [2]. Since BC often affects women of working age, it inevitably interferes with the earning [3,4,5] and employment opportunities [6] of those who survive breast cancer (breast cancer survivors, BCSs). Consequently, BC is not only the disease that causes the highest level of morbidity, mortality, and disability in women [7] but it is also the one that has the greatest economic impact [8].

The return to work (RTW) of BCSs is associated with a better prognosis [9] and quality of life [10]. For these reasons, medical research must encourage RTW. One of the few longitudinal studies available on BCSs, which observed a prolonged absence from work, loss of work ability, and residual disability in BCSs one year after surgery, suggested that a tailored policy in the workplace was needed to encourage RTW [11]. Unfortunately, to date, there have been no significant applications of such a policy anywhere in the world. Although the medical field has recognized the importance of integrated care in treating BCSs and improving their quality of life [12,13], the involvement of the occupational physician in providing ergonomic initiatives to facilitate the RTW of BCSs remains an exceptional measure [14], albeit one of recognized effectiveness [15,16].

In 2022, to fill this gap, we launched an initiative aimed at improving the link between hospital care and the work environment [17]. Women who had completed acute treatment and intended to return to work were assisted with a personalized intervention designed to examine their condition, residual health problems and disabilities in relation to occupational requirements and national policies on cancer survivors. Each patient was assisted by a team of occupational physicians who compared their clinical assessment with the work situation, which was studied in both physical and managerial aspects, analyzing the organizational justice perceived by each BCS and their job tasks. At the baseline, the occupational health characteristics of BCS workers (work ability, sleep, fatigue, emotional state) were compared with a group sized 5n of healthy female workers. Furthermore, a qualitative analysis was carried out of the factors that women considered important in the process of returning to work [17]. Each woman received advice on how to manage her health status in relation to work demands together with a letter for the doctor in charge of the health surveillance of workers. One year after this initial contact, at follow-up, we re-contacted the women to find out whether their reintegration into work had taken place smoothly and whether the advice given had been helpful.

The aims of this follow-up study were to check whether the factors that at the baseline had been postulated as favoring or hindering RTW had actually exerted this effect. Furthermore, we wanted to verify whether the occupational health conditions were the same as at the baseline or whether there had been improvements in the parameters measured one year earlier. Finally, we wanted to know whether the women felt the job recovery process was successful and whether they thought our intervention was helpful in promoting RTW. To achieve these objectives, we combined a qualitative analysis of the BCS responses to an interview with a quantitative comparison of the values obtained from standardized questionnaires relating to certain critical aspects such as sleep, fatigue, and mental balance, at the baseline and at follow-up.

## 2. Materials and Methods

### 2.1. Study Design

This study followed a prospective, observational design, with a non-pharmacological intervention, consisting of providing indications for recovering work. The self-selected sample was made up of women who, at the end of the hospital therapeutic course, requested assistance from the occupational medicine unit. The mixed methods included the use of (i) qualitative data on barriers and promoters of RTW, which were collected at the baseline with semi-structured interviews in person and were verified at the follow-up with telephone interviews; (ii) quantitative data on health conditions, which were compared at the baseline with a control group of women of the same age and at the follow-up with the values of the same subject at the baseline.

In 2022, at the baseline, 32 BCSs requesting assistance in returning to work (mean age 50.03 ± 8.99 years) underwent a medical examination conducted by one senior and three resident occupational doctors. In addition to an analysis of their clinical history (8 were classified as stage I, 12 as stage II, 8 as stage III, 4 as stage IV; 6 were undergoing chemotherapy, 24 hormone therapy) and an evaluation of their current condition (8 had lymphedema, 13 surgical sequelae, 21 musculoskeletal functional limitations), the aim of the examination was to study their previous work experience by analyzing their work tasks and studying the perception of organizational justice in the work environment that the worker had developed before the onset of the disease. Low perceived organizational justice has been shown to play an important role in the health and well-being of employees [18,19]; furthermore, the evaluation of the organization helps to understand the mechanisms through which it is possible to favor the reintegration of the worker. At the baseline, some characteristics of psychophysical health (sleep, fatigue, work ability, anxiety, depression, happiness) were measured using standardized questionnaires, and the data were compared with a 5 times larger sample of women of the same age without BC, employed in companies monitored by the occupational doctors who were performing this study. The BCSs were also invited to participate in a semi-structured interview designed to obtain detailed information concerning their working environment conditions and investigate the factors that hindered or could have favored their return to work. The qualitative data emerging from the survey were analyzed using thematic analysis. Other details on this first part of the study can be found in the relevant publication [17].

At the end of the first medical examination, the occupational doctors prepared a letter of advice for the doctor in charge of the worker’s health surveillance, aimed at improving the individual’s reintegration into her occupational environment.

This article reports the results of the follow-up that was conducted in 2023. One year after the first interview, the women were contacted again by telephone, for a short semi-structured interview in which they were asked if they had had a smooth RTW or if there had been any problems. On this occasion, the women were invited to indicate the factors hindering or favoring the process of reintegration into work, and to complete a set of questionnaires online that were identical to those used at the baseline.

An informed consent form outlining the objectives of this study and the purpose of the interview was given to each woman. The women were assured that their participation in this study was entirely optional and that they might leave the interviews at any moment without having to provide a reason. They were also assured that the survey was strictly confidential and anonymous. This project was approved by the Research Ethics Committee of the Università Cattolica del Sacro Cuore of Rome (project nr. 4672, approved 14 February 2022).

### 2.2. Questionnaire

The set of questionnaires included some standardized models.

Work ability was measured using the first question of the Italian version [20] of the Work Ability Index [21], which evaluates residual work ability compared to the maximum possessed during working life. This is known as the Work Ability Score (WAS). The convergent validity [22] and highly significant correlation [23] of this shortened version with the full questionnaire have been demonstrated. The WAS score varies from 0 to 10 and can be categorized into poor (0–5 points), moderate (6, 7), good (8, 9), and excellent (10) work ability [24].

Perceived organizational justice (OJ) [25] was measured using the Italian version [26] of the Colquitt questionnaire [27], which is composed of 20 questions that distinguish procedural (PJ), distributive (DJ), interpersonal (IntJ), and informational justice (InfJ). All items have a 5-point Likert scale, graded from 1 to 5. Consequently, the total score (OJ) ranges from 20 to 100 points. In this study, the reliability of the scale was 0.952. The subscale ranges and Cronbach’s alpha were as follows: PJ (7–35) 0.894; DJ (4–20) 0.908; IntJ (4–20) 0.922; InfJ (5–25) 0.879.

The quantity and quality of sleep were measured by means of the Italian version [28] of the 18-item Pittsburgh Sleep Quality Index (PSQI) [29]. This questionnaire enables individuals to make a self-assessment of sleep duration and its quality. The score for sleep quality ranges from 0 to 21; scores above 5 are considered indicative of poor sleep [30]. Cronbach’s alpha in this study was 0.856.

Perceived fatigue during work was measured using the 10-item Fatigue Assessment Scale (FAS) [31]. This scale is made up of 10 items graded from 1 to 5 and can consequently provide scores between 10 and 50. The score of 24 is considered the cut-off for defining clinical fatigue [32]. Reliability, measured by Cronbach’s alpha, was 0.885.

Anxiety and depression were measured using the Italian version [33] of the Goldberg scale [34] (GADS), composed of 18 binary questions that provide an anxiety score and a depression score. A worker with a score at the cut off for either scale (that is, 5 for anxiety or 2 for depression) has a 50% chance of having a clinically important disturbance; above these scores, the probability percentage rises sharply. In this study, Cronbach’s alpha was 0.883 for the anxiety scale and 0.892 for the scale relating to depression.

Happiness was measured using the Abdel–Khalek single-item scale [35], graded 0–10.

### 2.3. Interview

The semi-structured interviews covered the following points:What are your health conditions? What therapy are you undertaking? What are your current symptoms?Are you working at present?-If YES: When did you start again? What is your current job? Since you returned to work, how do you think your superiors have behaved? And your colleagues? Overall, do you feel that the process of recovering your previous job was complete and satisfying?-If NO: Why did you not go back to work?Did the occupational health advice you received help you in any way?

Telephone interviews were recorded and analyzed according to a modified version of the Braun and Clarke six-phase thematic analysis [36]. The design of the qualitative data adhered to the COREQ (consolidated criteria for reporting qualitative research) [37]. Medical and occupational histories were analyzed, leading to the inductive development, revision, and transcription of codes in an orderly database (phase 1). After that, peer debriefings were conducted in order to provide an initial data codification (phase 2) based on a comprehensive reading of each registered interview. Afterwards, the codes were combined to form larger themes after being clustered based on comparable and parallel discoveries. Phase 3 of the RTW experience themes covered both good and negative aspects, and the collaborative approach to repetitive analysis helped boost the data’s reliability. A thematic map was created and sub-themes pertaining to the primary topics were incorporated (phase 4), based on findings gathered in the literature describing the RTW of BC women and linguistic fragments acquired from interviews. Themes were then categorized into “barriers” (negative aspects) and “facilitators” (positive aspects) (phase 5). This study’s findings were compiled into a final report (phase 6). The sentences were translated from Italian to English at the conclusion of the examination.

### 2.4. Statistics

The variables we collected were tested for normality and their characteristics were described (mean, median, standard deviation, range). Scores for the questionnaires completed at the follow-up were compared with the data collected at the baseline using non-parametric tests (Wilcoxon matched pair signed rank test). A comparison of the scores of the non-parametric variables in the subgroups of the study sample was carried out using the Mann–Whitney U exact significance test. The correlation between the variables of interest was studied with Spearman’s rho. Statistical analyses were performed using IBM SPSS Statistics for Windows, (Version 26.0., IBM Corp., Armonk, NY, USA, release 15.0).

## 3. Results

A year after the first RTW interview, not all the women who participated in the research were working. In some cases, a new pathology or complications had occurred that required new treatment. In other cases, serious family problems, such as the need to care for a close family member, had compelled BCSs not to return to work. Of the 32 women screened in 2022, 26 were working. All of them were invited to complete the online questionnaire, and 17 accepted.

### 3.1. Quantitative Data

The quantitative data are summarized in Table 1.

After resuming work, the BCSs’ opinions regarding the correctness of their work environment (OJ) did not change significantly (Colquitt’s score 70.1 ± 23.6 vs. 69.2 ± 14.8, related-samples Wilcoxon signed rank test *p* = 0.286) from those recorded at the baseline; however, women who had returned to work generally made a more positive assessment of the organizational justice of their workplace than they had at the baseline; only two of the BCSs indicated a worsening of their working climate, mainly due to a lack of interpersonal justice. Female workers who judged their RTW process positively reported a significantly higher level of organizational justice than other BCSs (77.1 ± 19.1 vs. 37.3 ± 8.2; independent-samples Mann–Whitney U exact significance test *p* = 0.006).

The BCSs who returned to work judged their working capacity more positively than they did before returning to work (WAS score 6.5 ± 3.4 vs. 4.9 ± 2.8); the increase approached statistical significance without reaching it (Wilcoxon signed rank test *p* = 0.058).

One year after returning to work, BCSs still had a high frequency of sleep problems. The average sleep duration was 6.8 ± 1.6 h (range 4–9.5 h). Sleep quality, measured by the PSQI, was 8.2 ± 4.6 points (range 2–15). Only 15% of women reported a sleep quality that could be defined as “good”, i.e., less than 5 points on the PSQI. Sleep quality did not improve compared to the baseline. On the contrary, the quality of sleep was worse than that of the same women at the baseline (8.2 ± 4.6 vs. 6.2 ± 2.6), even if the difference did not reach statistical significance (related-samples Wilcoxon signed rank test *p* = 0.065).

Fatigue measured with the FAS had an average score of 24.5 ± 9.2 (range 10–41). The average score was higher than the cut-off, indicating excessive fatigue. More than 70% of the women who returned to work had chronic fatigue. Among BCSs, fatigue was slightly less than that reported at the baseline (FAS score 24.5 ± 9.2 vs. 26.4 ± 9.1), but the difference was not significant (related-samples Wilcoxon signed rank test *p* = 0.277).

Anxiety in the sample had a mean score of 5.7 ± 2.9. Over 60% of women had a score above the cut-off and would probably have been diagnosed as “anxious” by a psychiatrist. The average score was higher than that measured at the baseline (5.7 ± 2.9 vs. 4.8 ± 2.6), but the difference was not significant (related-samples Wilcoxon signed rank test *p* = 0.385). The depression score was 4.1 ± 3.1. Almost 70% of women were depressed. Depression symptoms had increased from the baseline (4.1 ± 3.1 vs. 3.3 ± 2.4), but the null hypothesis of invariance could not be rejected (Wilcoxon signed rank test *p* = 0.357). The happiness score remained unchanged compared to the baseline (6.7 ± 2.3 vs. 7.3 ± 1.4; Wilcoxon *p* = 0.173).

Perceived organizational justice in the workplace was inversely proportional to anxiety, depression, and poor sleep quality. Fatigue was directly proportional to anxiety, depression, and poor sleep quality. Age was not correlated with the quantitative variables (Table 2).

### 3.2. Qualitative Data

The thematic analysis of the data obtained from the interviews confirmed the three themes that had been observed at the baseline: personal elements, and factors linked to the company and to society. Each of these themes was composed of a series of subthemes, or factors. Person-related factors included physical problems, motivational blocks, and cognitive and neuropsychological problems, which acted as barriers, and work engagement, surgical breast reconstruction, and integrative treatments, which were facilitators of RTW. Company-related factors included several subthemes hindering RTW (work overload, work underload, environmental and ergonomic factors, inadequate shifts) and others facilitating it (policies for RTW, ergonomic and schedule adjustments, social support from colleagues and superiors). Society-related factors included unequal access to welfare benefits and family conflict, which were seen as obstacles, and legal and welfare benefits for workers with cancer, telecommuting or teleworking, and social support from family members, which were seen as promoters of RTW. Overall, therefore, each of the three themes (person, company, society) involved negative or positive factors. Only when the three classes of factors simultaneously exerted a positive effect was the RTW complete and totally satisfactory (Figure 1). One year after RTW, only 19 of the cohort members (59.4%) said the process had gone well. The occupational doctor’s intervention aimed at resolving the critical issues arising in each individual case.

#### 3.2.1. Person-Related Factors

When they resumed work, almost all BCSs were on hormone treatment, often with induced menopause. Many of them manifested collateral symptoms (hot flushes, headaches). About one-third of the women underwent oral chemotherapy treatment, with frequent side effects that were often gastrointestinal or, more rarely, cutaneous (urticarial rash) or neuropathic. Two workers were awaiting surgical interventions (breast reconstruction, reoperation due to breast prosthesis dislocation). In addition to these iatrogenic symptoms, the most common problems were frequent fatigability, asthenia, and excessive fatigue. Bone, joint, and muscle pain and lymphedema were present in about one-tenth of the sample. Sleep quality was generally poor, and anxiety and depression were very common. A previous investigation made it possible to identify some health-related factors that were obstacles to work. These included physical problems (pain, fatigue, lymphedema) as well as motivational blocks and cognitive and neuropsychological problems (reduced concentration, decreased performance, apathy). Residual symptoms at the end of treatment constituted a significant obstacle to a full return to work, especially for women who performed manual tasks. A saleswoman from an appliance shop reported:


*“The lymphedema and the pain in my hand have made it very difficult for me to move objects and put them up on shelves. I work with the constant fear that my fingers will get stuck and tools will fall out of my hand.”*


In addition to the disorders attributable to BC, approximately one-quarter of the BCSs had manifested other pathologies, including musculoskeletal disorders (carpal tunnel syndrome, scapulohumeral periarthritis) and neoplastic diseases (polyposis of the uterus, primary lung tumor, primary tumor of the colon) unrelated to BC. One of the patients, with previous diagnoses of multiple chemical sensitivity syndrome and asthma, had shown a significant worsening of symptoms after undergoing antiblastic therapies. About one-fifth of the workers showed cognitive and neuropsychological problems (reduced concentration, worse performance, apathy, memory lapses) and insomnia. Motivational blocks were also reported:


*“I currently have a lot of difficulty doing normal, common daily activities; there are days when I feel a complete refusal to do anything.”*


Mental health was often at stake:


*“I haven’t gone back to work because the very idea gives me an anxiety attack. I’m afraid I’m having a nervous breakdown.”*


On the other hand, in many cases, excellent surgical and medical treatment facilitated the return to work. Surgical breast reconstruction had often provided very satisfactory results, even improving body image.


*“The results of the treatment were excellent. I resumed work sooner than expected.”*


Work engagement was also of great importance. Female workers were often keen to highlight the benefits of returning to work.


*“Work helped me a lot to recover my independence.”*


#### 3.2.2. Company-Related Factors

In the process of returning to work, company policies involving ergonomic and schedule adjustments, social support from colleagues and superiors, and other organizational measures were of great importance. Positive experiences were prevalent:


*“Both my colleagues and my direct superior helped me with everything possible.”*


Unfortunately, company-related policies are not always positive. Sometimes, the intention to protect the worker resulted in isolation and demotion. On other occasions, rather than reorganizing a job, managers simply excluded the worker from some tasks, implicitly placing a greater burden on colleagues who were called upon to carry out the assignment no longer entrusted to the worker. This did not encourage good relationships between workers:


*“I had a bit of trouble. I was not allowed to return to my previous job and was given a lower-level task. In my absence, my colleagues had taken my place. After many years of service, I was no longer taken into consideration. Others did not accept my illness and its concomitant disabilities. There are colleagues who complain that there are too many sick employees in the same work environment. It’s demoralizing that they no longer allow you to do what you’ve done all your life because of an illness.”*


At other times, the company was explicitly against any adaptation, and when the worker returned, she was expected to perform the work she did before becoming sick.


*“When I returned to the company, the human resources manager asked me why I had gone to the hospital to be examined by the occupational doctors, stating that the doctors defend themselves by requesting an excessive number of precautions, and therefore, if I had presented their indications, I would have been fired.”*


It is worth noting that the company that claimed it was unable to relocate a woman treated for breast cancer is not a small or very small company belonging to the 97% of Italian companies in this category but is a global giant employing, in our country alone, over 9000 people. Moreover, another BCS confided to us that, in the intervening period, she had taken and passed a public exam for a managerial role in an important state-controlled company with tens of thousands of employees but had been rejected in the pre-employment medical examination stage when she declared her illness.

A sizeable number of BCSs had to face a series of obstacles, overload (including frequent travel, excessive hours, new responsibilities) or underload (exclusion from tasks that they performed before and which were assigned to other colleagues), inadequate shifts (night shifts, long hours), and environmental and ergonomic factors. The numerous examples of this kind included the following:


*“I was allowed to do my work from home. However, because I didn’t go into the office for months, I felt a bit left out. Before I was diagnosed with breast cancer, I thought I would have a career, which has now ended. I am cut off from progress projects, initiatives, and training projects as well as trips and missions. They don’t take me into consideration. I’m not sure if I would have had a career without breast cancer, but the company didn’t help me prove my reliability!”*


Often, the company did not respond promptly to a worker’s needs but intervened after occupational health consultancy:


*“I had a somewhat traumatic return because I had ongoing problems. When I got your letter, I turned to the corporate “ethics officer” who took charge of my situation.”*


In some cases, difficulties did not appear immediately after resuming work but manifested themselves at a later stage. For example, it is well known that each BCS undergoes a series of check-ups for a period of five years. In order to attend these check-ups, workers must necessarily be absent from work. This may place them in a difficult situation, because although they are ill, they are not at home and therefore cannot respond to any appointments for medical control on the part of social security institutions, meaning that their employer may be asked to intervene.


*“The manager asks for a home visit to ascertain illness every time I take sickness leave to do my check-ups, even if I inform the company of the need to carry out a check-up on a certain day.”*


Very often, the worker was removed from her tasks and, in fact, demoted.


*“They placed me on the company toll-free number, I was practically demoted, but on the other hand I can’t do everything I did before at full speed.”*


#### 3.2.3. Society-Related Factors

Social support from family members can be an important factor in facilitating a return to work, but family problems can present a strong obstacle:


*“My real problems are family ones. My husband took my children away from me and hasn’t let me see them for 3 months.”*


Italian law provides numerous legal and welfare benefits for workers with cancer. However, some workers have pointed out that there is unequal access to these benefits since only workers with a permanent contract can take advantage of them without losing their job. For those with a fixed-term contract, the mere declaration of a chronic pathology can be a reason for not renewing their post. Other workers reported that asking for benefits such as exclusion from travel, missions, night work, or extended hours would damage their career prospects.

The recent pandemic led to national measures in favor of vulnerable workers, including BCSs, who were given the right to telework. This transitory and pandemic-related solution offered BCSs some advantages but also contained some pitfalls, because it made it difficult for them to keep their role without coming in contact with colleagues and subordinates. Furthermore, many companies maximized the use of teleworking during the pandemic, only to hastily return to in-person work as soon as the pandemic was over. This often produced problems for BCSs.


*“I would like to go back to the office. I am the head of an operational structure of 50 people, so I have many responsibilities on my shoulders. I don’t know if anything will happen to my job when I return. If I had to drive to go to work in-person, it would take me an hour plus an indefinite amount of time looking for a parking space, all time I save by staying at home. Staying at home enables me to manage my family and my fatigue. Even if I work from home a lot and am exhausted at the end of the day, I can organize my breaks better. During my lunch break, I can rest for 20 min in a much more comfortable environment than in the office, where it wouldn’t be possible to just lie down on the sofa.”*


Sometimes, the availability of legal measures designed to protect women is used against them, for example, by imposing an unwanted transfer to a location close to the address of residence, or by requiring the worker to undergo a medico-legal assessment of their work capacity. Unfortunately, BCSs frequently reported this type of abuse.

One sector in which BCSs are not effectively protected is the financial sector. Some of them said they had been warned that it was not appropriate to ask to take out a mortgage to buy the family home; it would be better if the request were made only by the husband. This discrimination can cause even greater damage if the BCS also decides to divest herself of ownership of the family home or of the other assets for which she requests financing.

### 3.3. Occupational Health Consultancy

Most cohort members (27/32, 84.4%), including all those who succeeded in returning to work, expressed a favorable opinion of the intervention. According to the workers, occupational health consultancy was useful mainly for providing training about their condition in relation to work:


*“I came to you for consultancy to obtain information regarding legal protection for BCSs. It helped me not only from a personal point of view but also from a corporate point of view because I showed the letter you sent me to my company doctor and, thanks to your advice, was able to telework. I must say that having someone to shield you in some way is a very positive experience since you feel that you are not alone.”*


However, only a few of the BCSs used consultancy for the purpose for which it was intended, i.e., as an indication for the company doctor. Some workers said that the consultancy was useful from a personal point of view, but they did not feel the need to show it to the company doctor as their employers had already taken the necessary steps.

Some workers said that if they had shown the consultancy to their company, they would have been fired.


*“The occupational health consultancy that you gave me was of great help from a personal point of view due to the indications I received; I felt that someone had listened to the symptoms that I never imagined I would have in my life. I can’t say the same from a business perspective as the company was deaf to my needs.”*


Despite the various difficulties experienced in their different companies, all the respondents were pleasantly surprised to have been contacted. They expressed an urgent need to be heard and to receive more attention given the numerous problems that arise during chronic illness. A woman who has overcome the acute phase of a disease is justly concerned about her health. With each new symptom, the question arises as to whether it could be a recurrence of the disease, an effect of the treatment, or a related event of another kind. Very often, it is not easy to answer this question. It is also difficult for her to know what kind of workload she can tolerate and, conversely, what could be harmful. The return-to-work examination, which is provided for by law after long-term absences, cannot solve this type of problem.


*“A lot is done in prevention and treatment, but in my chronic condition I feel rather isolated.”*


## 4. Discussion

### 4.1. The Intervention

This study is the second part of a project aimed at promoting the recovery of the job positions of women with BC, through contact between the hospital and companies. A year after offering advice with the aim to improve the reintegration of the sampled women into their workplace, we contacted the interested parties again to verify the results. Thematic analysis confirmed that the three types of factors that can hinder or favor RTW were still present one year after returning to work. Quantitative analysis demonstrated that a year after returning to work, BCSs were still burdened by health problems such as sleep disorders, anxiety, and depression. The effort to resume occupational activities often resulted in excessive fatigue. However, women believed that their work capacity increased after RTW and, on the whole, they were satisfied with the occupational health counseling aimed at addressing individual physical, psychological, and work-related challenges. This result was expected based on the literature. Satisfaction with interventions that helped women cope during the RTW process was also expressed for other tailored measures introduced for BCSs [38,39]. A review of the literature concluded that physical and multifunctional interventions tailored to the needs of the patient increased the rate of RTW of people with cancer [40].

Almost all the women in our sample felt that the counseling was useful for informing them about the evolution of the disease and the work factors that might interfere with the disease. Many women established a contact between the university hospital occupational doctors and their general practitioner or the doctor in charge of their company health surveillance. In these cases, there was a rapid and full RTW that gave workers great satisfaction. Unfortunately, however, this did not always happen, especially due to problems with company policies. Some women preferred not to present the consultancy report because they were afraid the company’s reaction would put their job at risk. Although our survey was conducted on a very limited sample, it demonstrated that the protection of workers with cancer is not universal, even in a country such as Italy that has a highly developed economy and comprehensive legislation for the protection of disabled persons.

### 4.2. Quantitative Analysis of Health

Our study has shown that BCSs return to work with a burden of physical problems that persist after therapy. At the medical examination carried out in 2022, at the baseline, 65.6% had pain, 25% lymphedema, and 40.6% surgical sequelae [17]. In the previous literature, physical constraints were identified as one of the barriers to RTW [41,42,43]. A patient’s quality of life and RTW are significantly impacted by pain, particularly in the axilla and arm on the afflicted side [44,45]. In addition to being a direct result of the illness, pain can also result from radiation, chemotherapy, or surgery [46]. After one year, we observed that the problems had diminished in our sample. In fact, in recent years, more targeted treatment and technological progress in the medical field have reduced the frequency of these side effects [47]. Furthermore, personalized rehabilitation treatment that employs a range of methods (such as lymphatic drainage [48,49] and on-land [50] or aquatic exercises [51]) can lessen patient discomfort considerably and improve work readiness. However, these services are not always available, and many BCSs express their frustration at the lack of rehabilitation services for their condition and needs [52]. Research has shown that ergonomic interventions can help facilitate RTW [53]. Close coordination between the occupational physician and company management are needed to implement these interventions in the workplace. In this project, our objective was to enhance this kind of communication so that every patient received the essential care they needed.

In our sample, at the baseline, more than 70% of participants reported feeling tired. Excessive fatigue (FAS score > 24) was present in 74.2% of BCSs but only in 25% of female workers of the same age, with an odds ratio of 8.63 (95% confidence interval 3.57; 20.84) [17]. One year later, we observed a slight and non-significant reduction in fatigue. For BCSs, cancer-related fatigue is the most prevalent and troubling symptom [54,55,56,57]. It frequently prevents them from going back to work. A study showed that, during a two-year follow-up period, fatigue was identified as a chronic problem for 24 percent of BCSs [58]. An increasing body of research points to an inflammatory origin for the fatigue, which is closely associated with changes in the immunological and neuroendocrine systems [59]. Nonetheless, there has also been evidence of a link with other variables. Weight fluctuations, menopausal symptoms, coping mechanisms, social support, and metabolic alterations in BCSs are all frequently associated with fatigue [60]. Pain and depression [61], sleep difficulties, emotional symptoms, and neuromuscular fatigability were found to be the most significant predictors of chronic fatigue in cancer patients [62].

A problem that probably attracts less attention, but is of considerable importance, concerns sleep deprivation, which is typical in women with BC. In 2022, 84.4% of the cohort had a PSQI score >5, indicating poor sleep quality, with an odds ratio of 2.29 (CI95% 0.83, 6.30) compared to controls [17]. One year after returning to work, 85% of the BCSs in our sample still had poor sleep quality and a short sleep duration. According to earlier research, BC patients frequently experience shorter sleep durations [63] and frequent sleep issues (difficulty falling asleep, nocturnal awakenings, and non-restorative sleep) such as insomnia [64]. They also manifest heightened symptoms of sleeplessness with no discernible improvement more than a year after diagnosis [65,66,67]. This can negatively affect their quality of life even years after treatment [68]. Cancer patients’ sleep disturbances may be associated with chronotype and circadian parameters [69].

Sleep disorders in BCSs are closely linked to psychosocial stress resulting from the disease itself, its treatment, and also the work environment. The literature has highlighted the etiopathogenetic role of iatrogenic factors (e.g., endocrine therapy [70], breast surgery [71], and chemotherapy and/or radiotherapy [72,73,74]) and the psychological impact of a breast cancer diagnosis (e.g., depression, anxiety, fear of recurrence [75]), as well as the possible role of host–tumor interaction molecular processes [76]. Occupational factors may also play a role. Our study highlights that low perceived organizational justice may be a factor associated with poor sleep. Furthermore, it seems that sleep disturbance and cancer are related to one another, as cancer seems to encourage disturbed sleep and insufficient sleep encourages carcinogenesis and the advancement of cancer [76]. Another of our studies indicates a close relationship between the level of organizational justice perceived by the worker before the onset of the illness and the effectiveness of the RTW process [77]. Occupational stress factors and stress-induced lifestyles have often been associated with BC [78]. It has been hypothesized that work-related factors may influence the survival rate of BC patients [79]. However, no association between occupational stress and breast cancer has so far been demonstrated by meta-analytic studies [80].

Fatigue and sleep issues are linked to both anxiety and depression. In our sample, most subjects were depressed and anxious. At the baseline, before RTW, we had observed that 66% of women with BC suffered from anxiety, 85% from depression, and 72% were unhappy. The odds ratios compared to the controls were 10.8 (CI95% 4.6; 25.3) for anxiety, 13.8 (CI95% 5.0; 38.1) for depression, and 2.8 (CI 95% 1.2; 6.4) for unhappiness [17]. After one year, levels of anxiety, depression, and unhappiness had not significantly changed. These data correspond to what appears in the literature. According to published research, between 11% and 16% of BCS patients have mixed symptoms of depression and anxiety, and anxiety connected to cancer is linked to an increased risk of sleep disturbances [81]. Mental health issues represent a significant obstacle for RTW. In previous studies, anxiety [82] and depression [83,84,85] were significantly associated with not going back to work. Changes in the hierarchy of life values due to BC are very important in the RTW process [86].

In our sample, the work ability reported by BCSs at baseline was about half of what they believed it had been before the illness. One year after returning to work, women showed a non-significant trend towards improvement in self-assessed work ability. This trend was expected based on the literature. Previous studies demonstrated that BC leads to at least one period of sick leave in the year following the diagnosis, and this causes costs estimated at between 10,000 [87] and over 30,000 EUR [3] per patient. In cases with an unfavorable clinical course, BC progression leads to a reduced likelihood of employment, an increase in the number of workplace hours missed, and an elevated cost burden [88]. Our sample included many women with a favorable course, for whom the disease was under control, but they nevertheless encountered considerable difficulties. All the problems found in our sample were present in previous literature studies that had focused on one of the specific factors. Loss of work-related productivity due to treatment is generally high. Chemotherapy after returning to work is associated with presenteeism costs comparable to those of absenteeism [89]. The problems with sleep and the emotional state that we observed in our sample are also of the utmost importance for productivity. Sleep disturbance increased healthcare expenditure by 2% and absenteeism by 8% [90]. Depression and a history of anxiety were associated with lower work ability in Australian BCSs [91]. According to an Australian study, productivity loss following return to work (RTW) was around four times higher in the depressed group than in the non-depressed group [91]. In accordance with what we found, a longitudinal Chinese study conducted within a year of RTW noted that BCSs showed significant productivity loss and activity impairment [11]. These data are universally known and explain why women who survive BC have difficulty keeping their jobs or finding work. Despite the numerous laws designed to protect BCSs, our case study shows that discriminatory behavior towards BCSs is present in both small and large companies.

### 4.3. Qualitative Analysis of the RTW Process

The experiences gathered indicate that the RTW process is not always simple or free of obstacles. The health conditions of BCSs inevitably reduce work ability. Support measures, including those related to welfare and family, are not always effective. The company is the most critical setting where an incorrect approach can cause serious damage. The results of this study lead us to believe that companies that reinstate BC patients expect some kind of incentive. For example, in the first year of RTW, the State could make 50% of the salary paid to BCSs exempt from tax because a self-assessment of BCSs indicated that their illness had reduced their previous working capacity by approximately one half. In countries such as Italy where employers are obliged to pay an insurance premium for the risk of accidents and occupational diseases, insurance companies could exempt employers from paying this premium for BCSs.

Consideration should also be given to another problem that is closely linked to the inevitable loss of work ability that arises as a result of cancer. In cases where the company is unable to effectively adapt work tasks, a portion of the assignments previously carried out by the worker are dumped on other workers, thus creating possible tension between colleagues. In conflicts, the weakest succumbs. Discrimination against disabled individuals, or ableism, is a condition that is becoming more common in the workplace [92]. Even though programs and rules have been created to encourage companies to handle disability in an appropriate way [93,94], they have not always reached their objectives. Reduced work ability is a significant predictor of workplace violence and bullying [95]; therefore, BCSs may be exposed to unfair treatment and harassment more frequently than healthy colleagues.

Although our country provides universal and free healthcare, BC is always associated with economic losses for patients. This problem has been well documented in the literature. BC patients of working age face financial losses due to missed work and missed home productivity [96]. Minorities are disproportionately exposed to the economic problems associated with cancer [97,98]. A quarter of women with BC experience a reduction in income that is present even one year after returning to work [99]. Studies reported that non-manual workers may undergo the highest loss of productivity [100]. Moreover, unwanted work-related outcomes are more likely to occur for blue-collar workers who have a lower household income, worse financial well-being, lower RTW self-efficacy, worse job satisfaction, worse illness perception, more physical symptom distress, impaired physical functioning, and unfavorable work conditions [11]. In Italy, almost 56,000 new cases of BC were recorded in 2023 [101]. Considering that a working woman earns on average 16,550 EUR per year in our country [102], and that the employment rate for women in Italy is close to 50% [103], the income produced annually by female workers affected by BC is equal to 463 million EUR per year, which corresponds to 2% of the State Budget Law. These figures show the economic impact of BC and the damage it can cause to the economy. Furthermore, in addition to the drop in income caused by the disease, we observed that a woman with BC has difficulty accessing credit and is therefore sometimes induced to divest herself of the assets she owns. To overcome this type of problem, a law has recently been passed in Italy that allows ‘oncological oblivion’, i.e., without breaching the contract, oncological patients can avoid declaring their pathology to bodies that provide finance; however, this measure only comes into force 10 years after recovery, and therefore does not concern the patients we observed.

### 4.4. The Role of Occupational Health Services

An effective RTW for women with BC is associated not only with a better quality of life and greater survival [10,104], but also with clear beneficial economic effects for the individual, the company, and society. For this reason, it is important that clinical care be integrated with occupational medical measures. Hospital specialists should assist occupational doctors and companies in choosing measures that can promote a speedy recovery of work ability. Ergonomic intervention in the workplace should be accompanied by organizational measures that can reduce fatigue and sleep deprivation. Employers agree that providing tailored support to BCSs is a necessary measure to encourage RTW [105].

When BCSs return to work after suffering from fatigue, they require assistance in recovering from both physical and mental work-related exhaustion and in reaching a regular sleep schedule. This kind of intervention would bring an economic advantage as it would enhance BCSs’ work capacity. Since poor sleep is linked to mental health problems, stress management and good sleep hygiene should go hand in hand. It is also important to note that a low degree of perceived justice in the workplace is directly correlated with poor sleep quality. Therefore, the employer’s primary responsibility to BCSs is to enhance their work organization.

Clearly, an occupational doctor who is an expert on labor problems plays a vital role in assisting this process. A Swedish study demonstrated that BCSs who received advice and support on occupational matters or were encouraged to return to work had fewer sickness absences [106]. An efficient return to work depended largely on the nature of meetings with doctors; positive respectful encounters significantly reduced absenteeism in BCSs [107].

Women who have been successfully treated for BC still need medical assistance, which is not currently provided by the hospital or the general practitioner. Prevention principles and the numerous benefits for workers with disabilities need to be carefully evaluated and applied rationally, so that they do not impede the resumption of healthy and safe work. Because of ambiguity in some welfare measures, a significant number of female workers in our sample did not apply to be registered as disabled or for legal benefits such as exemption from co-payment for health expenses, the possibility of working fewer hours, or taking days off for medical treatment, because they feared that these benefits might cause them to lose their job. A meta-review indicates that BCSs continue to face challenges and require interventions to address these difficulties [108,109]. Our research indicates that many more studies are needed to clarify which measures are essential for the safety of women who have had BC and how an RTW can be implemented in a way that does not jeopardize future career prospects and full job satisfaction.

### 4.5. Strengths and Weaknesses of This Study

Our study had the advantage of being, to the best of our knowledge, the only longitudinal study that combines a qualitative investigation of the obstacles and factors favoring RTW with the quantification of levels of physical and mental disability. It attempted to combine medical techniques, such as examining patients, with a psychological investigation of their needs and a technical analysis of working conditions. Many studies have examined health problems that can affect BCSs, such as poor sleep, fatigue, anxiety, depression, and poor work ability, but none have measured them simultaneously and compared them with the state of healthy workers. Furthermore, few projects have combined assessments of individual health with those of workplace safety and fairness. No project, as far as we know, has thought of creating a connection between the hospital from which the worker is discharged and the job to which she must return. Unlike other studies targeting a specific job sector or a pre-selected population, our project included all women who requested assistance. As regards the factors favoring or hindering RTW, the researchers did not start from a hypothesis to be tested, but accepted all the observations of the BCSs and organized them through thematic analysis. It was thus possible to observe that some topics that the literature has largely dealt with, such as the physio-psychological conditions of BCSs, company policies, and the benefits that society provides for cancer patients, can have an ambiguous effect, sometimes of benefit, and sometimes of an obstacle to RTW. The researchers were guided by the needs of BCSs and sought the best compatibility between their work possibilities and the environment in which they had to return to being productive.

However, this study has important limitations. The main one is that it was based on a self-selected sample of women who asked for assistance from the occupational doctor at the end of their hospital treatment. The sample therefore is not representative of all the women with BC treated every year in our hospital. Great caution is necessary in inferring the applicability of our results to the totality of BCSs. Nevertheless, this study has documented the numerous critical issues that the RTW process presents. Since the cohort included all women who requested assistance, randomization was not possible. Consequently, at the baseline, we compared the cohort data with those of an external sample of age-matched female workers not affected by BC, while at the follow-up, each BCS was its own control. This method could not prevent significant loss of cases at the follow-up. Doctor conversations are challenging, especially for women who have faced many medical procedures. It is not surprising that some BCSs have no interest in being interviewed again one year after the RTW. The number of cases examined, although greater than that of other qualitative studies, is not sufficient to verify the significance of the differences observed in the quantitative variables. A greater number of patients is certainly necessary to verify some of the hypotheses that seem to emerge from these observations, for example, that work ability tends to improve with the resumption of work and that fatigue progressively reduces. Sleep and mental health problems likely require more time and larger samples to register significant changes. A longer observation period could allow us to verify the possible recovery of the investigated parameters.

## 5. Conclusions

Our project, created to help women with BC return to work, has documented the existence of three types of factors, linked to the person, the company, and society, which can favor or hinder the process of recovering work activity. Some of the obstacles observed at the beginning of the process were still present one year after returning to work. Women reported a significant reduction in work ability, which was mainly associated with fatigue, altered sleep patterns, and the presence of symptoms of anxiety and depression. The personalized intervention designed to overcome obstacles and reintegrate women into their careers was greatly appreciated by BCSs. At the end of their hospital treatment, women with BC need a personalized care path to recover their working capacity and improve their satisfaction and quality of life. Companies must be assisted in designing a tailored set of interventions for BCSs, while the medical sector must help women struggling to overcome the problems of a chronic pathology.

## Figures and Tables

**Figure 1 ijerph-21-01057-f001:**
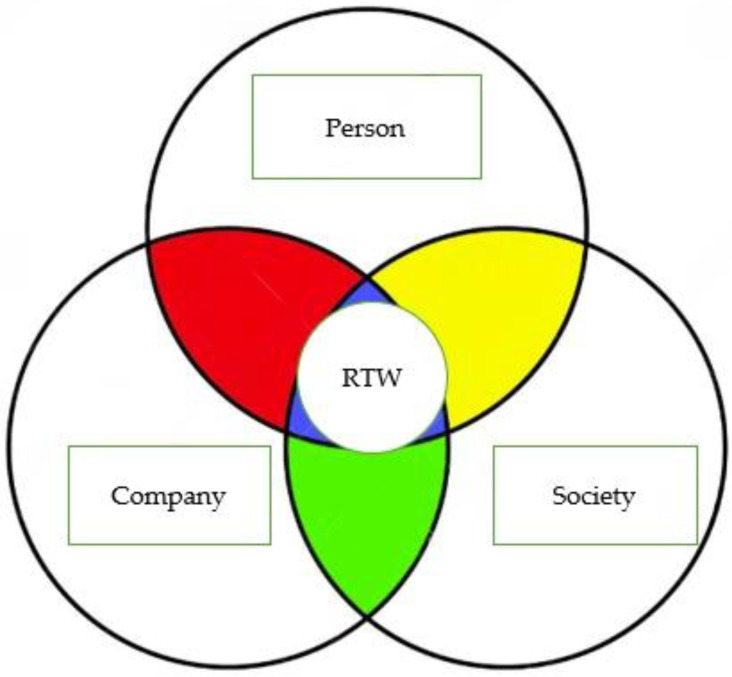
A complete and satisfactory return to work (RTW) requires the complementary efficiency of personal, company, and societal factors.

**Table 1 ijerph-21-01057-t001:** Quantitative data at follow-up compared with baseline.

Variable (Score Range)	Baseline (mean ± s.d)	Follow-Up (mean ± s.d)	*p*-Value *
Organizational justice (20–100)	69.2 ± 14.8	70.1 ± 23.6	0.286
Work ability (1–10)	4.9 ± 2.8	6.5 ± 3.4	0.058
Fatigue (10–50)	26.4 ± 9.1	24.5 ± 9.2	0.277
PSQI (0–21)	6.2 ± 2.6	8.2 ± 4.6	0.065
Anxiety (0–9)	4.8 ± 2.6	5.7 ± 2.9	0.385
Depression (0–9)	3.3 ± 2.4	4.1 ± 3.1	0.357
Happiness (0–10)	7.3 ± 1.4	6.7 ± 2.3	0.173

Notes: * Related-samples Wilcoxon signed rank test.

**Table 2 ijerph-21-01057-t002:** Correlations between the quantitative variables.

		Age	Justice	Anxiety	Depression	PSQI	Fatigue
Age	Spearman’s rho	1.000	0.063	−0.072	0.062	0.095	0.342
	Sig. (2-tailed)		0.736	0.695	0.735	0.605	0.060
Justice	Spearman’s rho	0.063	1.000	−0.549 **	−0.605 **	−0.445 *	−0.287
	Sig. (2-tailed)	0.736		0.001	0.000	0.012	0.118
Anxiety	Spearman’s rho	−0.072	−0.549 **	1.000	0.583 **	0.723 **	0.420 *
	Sig. (2-tailed)	0.695	0.001		0.000	0.000	0.019
Depression	Spearman’s rho	0.062	−0.605 **	0.583 **	1.000	0.595 **	0.579 **
	Sig. (2-tailed)	0.735	0.000	0.000		0.000	0.001
PSQI	Spearman’s rho	0.095	−0.445 *	0.723 **	0.595 **	1.000	0.521 **
	Sig. (2-tailed)	0.605	0.012	0.000	0.000		0.003
Fatigue	Spearman’s rho	0.342	−0.287	0.420^*^	0.579 **	0.521 **	1.000
	Sig. (2-tailed)	0.060	0.118	0.019	0.001	0.003	

Notes: * *p* < 0.05; ** *p* < 0.01.

## Data Availability

The datasets generated and analyzed during the current study are not publicly available due to the sensitive and identifiable nature of our qualitative data but are available from the corresponding author on reasonable request.

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
