# Peer review of "Returning to Work after Breast Cancer: A One-Year Mixed-Methods Study"

_ijerph, 2024, doi:10.3390/ijerph21081057_

Round 1

Reviewer 1 Report

Comments and Suggestions for Authors

The incidence of cancer is rising in many countries. Many of those affected are working people and the question of returning to work after acute treatment is still highly topical. In this respect, the study presented here deals with a very relevant topic in terms of health policy. Specifically, it evaluates an intervention with the aim of facilitating the return to work for breast cancer patients in Italy using a pre-post comparison.

Introduction

The introduction is brief, but makes the intention of the study and its relevance quite clear. What is missing, however, is a more detailed description of the intervention. As a reader, it was not clear to me how exactly the intervention was carried out, which makes it difficult for me to interpret the results. I suggest characterising the intervention in more detail either in the introduction or in the methods section.

Methods

The methods section lacks a description of the sampling! How were the patients recruited, where were they recruited, what was the response rate, was a case number calculation carried out, how many of the patients surveyed were actually employed before the illness? This information is absolutely essential.

Results

A classic Table 1 describing the sample is missing.

The evaluation of the pre-post comparisons is not shown in the results section. There is only information in the text. It would be very helpful to have a table presented here. I also miss the comparison between patients who had returned to work and those who had not. How do these two groups differ? In general, the quantitative results are presented much more concisely than the qualitative results. In the latter case, the text could even be shortened somewhat to emphasise only the core results.

Discussion

The study was not randomised, which limits its significance. This problem should be discussed in more detail. Another disadvantage that catches the eye is the very small sample size. Why was such a small number of cases chosen? What implications did this have for the analysis? Overall, the limitations of the study should be presented more transparently and their implications discussed.

Minor remarks

Line 35 following: the sentence is very convoluted and complicated to read.

Comments on the Quality of English Language

OK

Author Response

Reviewer#1

The incidence of cancer is rising in many countries. Many of those affected are working people and the question of returning to work after acute treatment is still highly topical. In this respect, the study presented here deals with a very relevant topic in terms of health policy. Specifically, it evaluates an intervention with the aim of facilitating the return to work for breast cancer patients in Italy using a pre-post comparison.

Response: We sincerely thank the reviewer for the attention with which he/she reviewed our manuscript, providing many useful suggestions for improvement. We understood that the follow-up nature of our study had not been well described and many references to the baseline study were missing. We have now included these data, to make it easier to understand even for those who have not read the first article.

Introduction

The introduction is brief, but makes the intention of the study and its relevance quite clear. What is missing, however, is a more detailed description of the intervention. As a reader, it was not clear to me how exactly the intervention was carried out, which makes it difficult for me to interpret the results. I suggest characterising the intervention in more detail either in the introduction or in the methods section.

R.: We thank the reviewer who highlighted an important weakness of the manuscript. In writing the text, in fact, we did not fully explain the method used and the results achieved at baseline. We have now inserted a new paragraph at the beginning of the methods section, which is as follows:

“This study followed a prospective, observational design, with a non-pharmacological intervention, consisting of providing indications for recovering work. The self-selected sample was made up of women who, at the end of the hospital therapeutic course, requested assistance from the occupational medicine unit. The mixed method included: i) qualitative data on barriers and promoters of RTW which were collected at baseline with semi-structured interviews in person and were verified at follow-up with telephone interviews; ii) quantitative data on health conditions which were compared at baseline with a control group of women of the same age and at follow-up with the values of the same subject.”

Methods

The methods section lacks a description of the sampling! How were the patients recruited, where were they recruited, what was the response rate, was a case number calculation carried out, how many of the patients surveyed were actually employed before the illness? This information is absolutely essential.

R.: The reviewer is right. All this information was part of the baseline study and must be summarized in this work. We have placed the information related to the first study in a methods paragraph, which now reads as follows:

“In 2022, at baseline, 32 BCSs requesting assistance in returning to work (mean age 50.03 ± 8.99 years) underwent a medical examination conducted by one senior and three resident occupational doctors. In addition to an analysis of their clinical history (8 were classified as stage I, 12 as stage II, 8 as stage III, 4 as stage IV; 6 were undergoing chemotherapy, 24 hormone therapy) and an evaluation of their current condition (8 had lymphedema, 13 surgical sequelae, 21 musculoskeletal functional limitations), the aim of the examination was to study their previous work experience by analyzing the work task and studying the perception of organizational justice of the work environment that the worker had developed before the onset of the disease. Low perceived organizational justice has been shown to play an important role in the health and well-being of employees [18, 19]; furthermore, the evaluation of the organization helps to understand the mechanisms through which it is possible to favor the reintegration of the worker. At baseline, some characteristics of psychophysical health (sleep, fatigue, work ability, anxiety, depression, happiness) were measured using standardized questionnaires, and the data were compared with a 5 times larger sample of women of the same age without BC, employed in companies monitored by the occupational doctors who were performing this study. The BCSs were also invited to participate in a semi-structured interview designed to obtain detailed information concerning their working environment conditions and investigate the factors that hindered or could have favored their return to work. The qualitative data emerging from the survey were analyzed using Thematic Analysis. Other details on this first part of the study can be found in the relevant publication [17].

At the end of the first medical examination, the occupational doctors prepared a letter of advice for the doctor in charge of the worker's health surveillance aimed at improving the individual’s reintegration into her occupational environment.”

Results

A classic Table 1 describing the sample is missing. The evaluation of the pre-post comparisons is not shown in the results section. There is only information in the text. It would be very helpful to have a table presented here. I also miss the comparison between patients who had returned to work and those who had not. How do these two groups differ? In general, the quantitative results are presented much more concisely than the qualitative results. In the latter case, the text could even be shortened somewhat to emphasise only the core results.

R.: Accepting the reviewer's request, we added a table containing all the data resulting from questionnaires. The pre-post comparisons do not reach significance, also because the size of the cohort is modest. For this reason, the quantitative data took on much less importance in this second part of the study than it had at baseline, when we compared the variables measured in the cohort with 160 controls.

At follow-up, there were six BCS who were not working; as we explained, some had had complications that prevented them from returning to work, others had obtained pensions. The group was too small to do a quantitative analysis. In contrast, the qualitative data collected at follow-up comes from a fairly large sample, larger in size than many qualitative studies on the same topic. For this reason, the exposition of qualitative data has gained greater space.

Following the reviewer's suggestion, we reduced the space dedicated to the analysis of the qualitative results and enlarged that of the quantitative data, also reporting the baseline results in various parts of the manuscript. In the Discussion we have separated the various parts into different sub-chapters, so as to make reading easier.

Discussion

The study was not randomised, which limits its significance. This problem should be discussed in more detail. Another disadvantage that catches the eye is the very small sample size. Why was such a small number of cases chosen? What implications did this have for the analysis? Overall, the limitations of the study should be presented more transparently and their implications discussed.

R.: We have extended the limitations section, although we must clarify that the study was a census of all those who in 2022, having to return to work, asked for assistance from the occupational medicine of the university hospital. These conditions do not allow randomization. The sample size calculation conducted before the start of the project had indicated a minimum number of 21 cases to observe quantitative differences in the parameters of interest; in fact, in the baseline we observed many differences compared to women without breast cancer. The size of the cohort was very large for a qualitative study and in fact it required a strong commitment from numerous researchers. The sub-section now is as it follows:

“4.5. Strengths and weaknesses of the study

Our study had the advantage of being, to the best of our knowledge, the only longitudinal study that combines a qualitative investigation of the obstacles and factors favoring RTW with the quantification of levels of physical and mental disability. It attempted to combine medical techniques, such as examining patients, with the psychological investigation of their needs and the technical analysis of working conditions. Many studies had examined some of the health problems that can concern BCS, sleep, fatigue, anxiety, depression, poor work ability, but none had measured them simultaneously and compared them with the state of healthy workers. Furthermore, few projects have combined assessments of individual health with that of workplace safety and fairness. No project, as far as we know, has thought of creating a connection between the hospital from which the worker is discharged and the job to which she must return. Unlike other studies targeting a specific job sector or a pre-selected population, our project included all women who requested assistance. As regards the factors favoring or hindering RTW, the researchers did not start from a hypothesis to be tested, but accepted all the observations of the BCSs and organized them through thematic analysis. It was thus possible to observe that some topics which the literature has largely dealt with, such as the physio-psychological conditions of BCSs, company policies and the benefits that society provides for cancer patients, can have an ambiguous effect, sometimes of benefit, sometimes of obstacle to RTW. The researchers were guided by the needs of the BCSs and sought the best compatibility between their work possibilities and the environment in which they had to return to being productive.

However, this study has important limitations. The main one is that it was based on a self-selected sample of women who asked for assistance from the occupational doctor at the end of their hospital treatment. The sample therefore is not representative of all the women with BC treated every year in our hospital. Great caution is necessary in inferring results on the totality of BCSs. Nevertheless, it documented the numerous critical issues that the RTW process presents. Since the cohort included the census of all women who requested assistance, randomization was not possible. Consequently, at baseline we compared the cohort data with those of an external sample of age-matched female workers not affected by BC, while at follow-up each BCS was its own control. This method could not prevent significant loss of cases at the follow-up. Doctor conversations are challenging, especially for women who have faced many medical procedures. It is not surprising that some BCSs have no interest in interviewing again one year after the RTW. The number of cases examined, although greater than that of other qualitative studies, is not sufficient to verify the significance of the differences observed in the quantitative variables. A greater number of patients is certainly necessary to verify some of the hypotheses that seem to emerge from these observations, for example that work ability tends to improve with the resumption of work and that fatigue progressively reduces. Sleep and mental health problems likely require more time and larger samples to register significant changes. A longer observation period could allow us to verify the possible recovery of the investigated parameters.”

Minor remarks

Line 35 following: the sentence is very convoluted and complicated to read.

R.: We have rewritten the period, which now is:

              “One of the few longitudinal studies available on BCSs, which observed prolonged absence from work, loss of work ability, and residual disability in BCSs one year after surgery, suggested that a tailored policy in the workplace was needed to encourage RTW [11].”

Reviewer 2 Report

Comments and Suggestions for Authors

Please see attached review

Comments on the Quality of English Language

Generally good but some aspects need re-wording as outlined in the review

Author Response

Reviewer #2

Return to work after breast cancer. A one-year mixed-method study.

Thank you for this interesting study on an important topic. There are however some aspects of the study that require greater clarity

Response: We sincerely thank the reviewer for the time he/she took to review our manuscript, indicating many points that needed improvement. We had not sufficiently described the results collected at the baseline and the methods followed in the first part of the research, and consequently we had not made the results of this second study clear. We have now corrected the manuscript in many places, summarizing the results of the first study, to allow all readers to understand the follow-up results.

Abstract

Line 19: Please clarify the meaning of “protection”

R.: The protection of workers with disabilities is provided for by many legislations, including ours. The meaning of the term protection is detailed in the discussion, at lines 420-425, and was reported with the same meaning in a patient's statement, at line 361. In the abstract, the meaning of the term is clarified in the sentence in which it is inserted: "The interviews demonstrated that, even in an economically developed country that has provided numerous benefits for BCSs, protection is not always effective”.

Introduction

Line 35: Please clarify the statement: “medical research must encourage RTW”

R.: We thank the reviewer for inviting us to explain in more detail the importance of recovering from work in women who have had breast cancer. In the first article we dedicated a page to this topic. In this follow-up article, we limited ourselves to a summary consideration that “The return to work (RTW) of BCSs is associated with a better prognosis [9] and quality of life [10]. For this reason, medical research must encourage RTW (lines 34-35).” We preferred to keep the introduction succinct, because the article already seems long enough to us and it did not seem necessary to repeat the arguments on the importance of breast cancer, because we imagine that readers are largely aware of it.

Lines 55-57: You need to re-word the aim/s of the study as they are currently not clear: “The aims of this study were to check whether, one year after returning to work, the factors that could hinder or favor the return the same as those expected at baseline and to evaluate the effectiveness of the intervention carried out”

R.: We absolutely agree. We considered it necessary not only to rewrite the sentence, dividing it into three periods so that it was clearer, but above all to explain the baseline results succinctly, so that the usefulness of the follow-up was clearer. We have rewritten the scope definition, which is now as follows:

“The aims of this follow-up study were to check whether the factors that at baseline had been postulated as favoring or hindering RTW had actually exerted this effect. Furthermore, we wanted to verify whether the occupational health conditions were the same as at baseline or whether there had been improvements in the parameters measured one year earlier. Finally, we wanted to know whether the women felt the job recovery process was successful and whether they thought our intervention was helpful in promoting RTW. To achieve these objectives, we combined a qualitative analysis of the BCS responses to an interview with a quantitative comparison of the values obtained from standardized questionnaires relating to certain critical aspects such as sleep, fatigue, mental balance, at baseline and at follow-up.”

Materials and Methods

This section is difficult to read as it lacks organisation. The Methods section should be structured according the standardised approach for reporting the methods for a study: Study design, sample (to include eligibility criteria, sample size etc) data collection methods and procedures, and data analysis.

R.: The reviewer is right. The text was disorganized, and the project description lacked a clear distinction between baseline and follow-up. Following the reviewer's advice, we described in the first lines the epidemiological design, the selection methods and the mixed methods used in the two phases. We have therefore reported, in a very short form, the results of the baseline study. Finally, we described the method of this study. The section now is as follows:

“This study followed a prospective, observational design, with a non-pharmacological intervention, consisting of providing indications for recovering work. The self-selected sample was made up of women who, at the end of the hospital therapeutic course, requested assistance from the occupational medicine unit. The mixed method included: i) qualitative data on barriers and promoters of RTW which were collected at baseline with semi-structured interviews in person and were verified at follow-up with telephone interviews; ii) quantitative data on health conditions which were compared at baseline with a control group of women of the same age and at follow-up with the values of the same subject at baseline.

In 2022, at baseline, 32 BCSs requesting assistance in returning to work (mean age 50.03 ± 8.99 years) underwent a medical examination conducted by one senior and three resident occupational doctors. In addition to an analysis of their clinical history (8 were classified as stage I, 12 as stage II, 8 as stage III, 4 as stage IV; 6 were undergoing chemotherapy, 24 hormone therapy) and an evaluation of their current condition (8 had lymphedema, 13 surgical sequelae, 21 musculoskeletal functional limitations), the aim of the examination was to study their previous work experience by analyzing the work task and studying the perception of organizational justice of the work environment that the worker had developed before the onset of the disease. Low perceived organizational justice has been shown to play an important role in the health and well-being of employees [18, 19]; furthermore, the evaluation of the organization helps to understand the mechanisms through which it is possible to favor the reintegration of the worker. At baseline, some characteristics of psychophysical health (sleep, fatigue, work ability, anxiety, depression, happiness) were measured using standardized questionnaires, and the data were compared with a 5 times larger sample of women of the same age without BC, employed in companies monitored by the occupational doctors who were performing this study. The BCSs were also invited to participate in a semi-structured interview designed to obtain detailed information concerning their working environment conditions and investigate the factors that hindered or could have favored their return to work. The qualitative data emerging from the survey were analyzed using Thematic Analysis. Other details on this first part of the study can be found in the relevant publication [17].

At the end of the first medical examination, the occupational doctors prepared a letter of advice for the doctor in charge of the worker's health surveillance aimed at improving the individual’s reintegration into her occupational environment.

This article reports the results of the follow-up that was conducted in 2023. One year after the first interview, the women were contacted again by telephone, for a short semi-structured interview in which they were asked if they had had a smooth RTW or if there had been any problems. On this occasion, the women were invited to indicate the factors hindering or favoring the process of reintegration into work, and to complete a set of questionnaires online that were identical to those used at baseline.”

The study aim includes “evaluate the effectiveness of the intervention”. However, a randomized control trial is required to achieve this aim but this is not the design of this study. Please revise the aim of the study

R.: We agree. There was a translation error, the term should have been "successfulness". We have corrected it, as above reported.

Study design

You need to describe the study design in this section. It appears that this was a mixed methods design but no information is provided on the design of the quantitative or qualitative aspects of the mixed methods study.

R.: As we said above, we rewrote this part.

Line 63: “BCSs requesting assistance in returning to work ….” Information is needed on the profile (demographic and cancer-related) of these women including the number of women who requested assistance. What was the eligibility criteria for inclusion in the study? Had some of the women in your sample already returned to work prior to the ‘intervention’ as it seems this way from some of the qualitative data? You need to clarify this. Greater detail is needed on the comparison group: “the data were compared with a 5 times larger sample of women of the same age without BC, employed in companies monitored by the occupational doctors who were performing this study”. You need to describe the women in this data base

R.: We agree. Confirming what we explained above, we rewrote this part.

Quantitative data

What was the primary outcome measure for this study? Was the study powered for the primary outcome measure?

R.: The sample size was self-determined by the women's request for assistance. All those who requested assistance were included and were contacted again one year later. Naturally, at baseline the sample size calculation necessary to highlight significant differences between women with pathology and healthy women was carried out. Based on this calculation it was decided to match the 32 women who requested assistance with a fivefold number of controls. Conversely, since the 2023 sample corresponds to the 2022 cohort, no sample size calculation was necessary. We are well aware of the fact that the size of the cohort has decreased, and this reduced the possibility of having significant differences. However, the purpose of the project was individual assistance, and this was independent of the number. We reported among the limitations of the study the small size of the cohort at follow-up and in the Discussion we recalled that inference is limited by this small number.

You need to explain “occupational justice” and its relevance for women with breast cancer with literature to support otherwise there is no clear rationale for including this measure. How was Fatigue measured? Describe the measurement tool used. You have reported Cronbach’s alpha for each of the measures. Were these calculated from your data? If so, this should be included in the Results section

R.: Also in this case, having dealt extensively with the meaning of Organizational Justice and the measurement of fatigue in the first article, we felt it was not necessary to repeat this description; however, we have cited the entire reference bibliography, which the reader can refer to for clarification. Accepting the reviewer's invitation, we have however expanded this description, succinctly indicating the reasons that determined the inclusion of this variable in the project We added 2 references. Cronbach's alphas have been reported next to the individual questionnaires to allow the reader to evaluate the reliability of the instruments before starting to read the research results. Most scientific journals, Including Health Care (MDPI) which published the first study, adopt this procedure.

Qualitative data analysis

Lines 134-136: You need to a detailed description of the methods used for analysis of the qualitative data. Explain “modified version” of Braun and Clarke. Discuss the strategies used to ensure a rigorous process was used to support the qualitative findings

R.: Accepting the reviewer's request, we have included a description of the thematic analysis which follows that already exposed in the previous work. The text now is as follows:

“Telephone interviews were recorded and analyzed according to a modified version of the Braun and Clarke six-phase Thematic Analysis [36]. The design of the qualitative data adhered to the COREQ (Consolidated criteria for reporting qualitative research) [37]. Medical and occupational histories were analyzed, leading to the inductive development, revision, and transcription of codes in an orderly database (phase 1). After that, peer debriefings were conducted in order to provide an initial data codification (phase 2) based on a comprehensive reading of each registered interview. Afterwards, the codes were combined to form larger themes after being clustered based on comparable and parallel discoveries. Phase 3 of the RTW experience themes covered both good and negative aspects, and the collaborative approach to repetitive analysis helped boost the data's reliability. A thematic map was created and sub-themes pertaining to the primary topics were incorporated (phase 4), based on findings gathered in the literature describing the RTW of BC women and linguistic fragments acquired from interviews. Themes were then categorized into "barriers" (negative aspects) and "facilitators" (positive aspects) (phase 5). The study's findings were compiled into a final report (phase 6). The sentences were translated from Italian to English at the conclusion of the examination.”

Results

Line 147: You state “A year after the first RTW interview, not all the women who participated in the 147 research were working. In some cases, a new pathology or complications had occurred that required new treatment. In other cases, serious family problems, such as the need to care for a close family member, had compelled BCSs not to return to work” You must give exact numbers for the highlighted cases.

R.: We had already indicated this number [line 151 of the old version].

Reasons why six women did not complete the questionnaires? What strategies were used by the team to obtain questionnaires from these six women?

R.: The informed consent document that all the workers signed stipulated that they could withdraw from the project at any time. All women were repeatedly solicited, but we did not implement any form of coercion for those who did not intend to respond. Taking the reviewer's suggestion, we have discussed this issue in the Discussion where we address the limitations of the study.

Line 160: “Female workers who judged their RTW process positively” – how was this measured?

R.: We thank the reviewer because he/she pointed out an element that was missing in the text. We classified the path as positive or negative based on the results of the interview. The question that was asked to all women during the telephone interviews was: "Overall, do you feel that the process of getting back to your previous job was complete and satisfactory? We have added this specification in the text.

Line 174: what is the cut-off for severe fatigue for the MAF – include this in the Methods section.

R.: A score ≥24 has been proposed as a cut-off for classifying fatigue on the FAS. This information had already been reported on line 114-115 of the manuscript, in the indicated section.

The women in this study have extremely high levels of anxiety and depression which were significantly higher at follow-up. You need to provide reasons for this in the Discussion section.

Line 184: happiness score remained unchanged compared to the baseline (6.7±2.3 vs. 7.3±1.4; Wilcoxon p=0.173). This is very odd given the very high prevalence of depression and anxiety. What do you think is the reason for this (include in Discussion)

R.: We thank the reviewer who allowed us to explain this point better. Follow-up results cannot be discussed without taking into account baseline conditions, which were analyzed and discussed in the first work. In this article we deemed it necessary, following the reviewer's advice, to briefly report the baseline situation. 66% of women with breast cancer suffered from anxiety, 85% from depression, 72% were unhappy. The odds ratio compared to controls was 11 for anxiety, 14 for depression, 3 for unhappiness. A year later, levels of anxiety, depression and unhappiness had not significantly changed.

Following the reviewer's suggestion, we have divided the Discussion into several sub-chapters, so as to make reading easier.

Qualitative Data

What were the subthemes in each of the three categories?

R.: Accepting the reviewer's invitation, we summarized the main sub-themes of the three categories that had been reported in the previous work. The text now is as follows:

“The thematic analysis of the data obtained from the interviews confirmed the three themes that had been observed at baseline: personal elements, factors linked to the company and to society. Each of these themes was composed by a series of subthemes, or factors. Person-related factors included physical problems, motivational blocks, cognitive and neuropsychological problems which acted as barriers, and work engagement, surgical breast reconstruction and integrative treatments which were facilitators of RTW. Company-related factors included several subthemes hindering RTW (work overload, work underload, environmental and ergonomic factors, inadequate shifts) and other facilitating it (policies for RTW, ergonomic and schedule adjustments, social support from colleagues and superiors). Society-related factors included unequal access to welfare benefits and family conflict which were seen as obstacles, and legal & welfare benefits for workers with cancer, telecommuting or teleworking, and social support from family members, which were seen as promoters of RTW. Overall, therefore, each of the three themes (Person, Company, Society) involved negative or positive factors. Only when the three classes of factors simultaneously exert a positive effect was the RTW complete and totally satisfactory (Figure 1). One year after RTW, only 19 of the cohort members (59.4%) said the process had gone well. The occupational doctor’s intervention aims at resolving the critical issues arising in each individual case.”

Line 202: “When they resumed work, almost all BCSs were on hormone treatment” Exactly how many were on hormone treatment?

R.: We have gladly added the numbers in the description of the cohort, as above reported.

This section is very long – you need to reduce the number and length of quotes.

R.: We willingly accepted this invitation; in fact, the quotes were too many and often repetitive. We have substantially reduced the qualitative section of results.

There are no clear qualitative findings related to participants’ experiences/perceptions of the intervention.

  1. Following the reviewer's request, we have reported the numbers. The response of the BCS to the question on the usefulness of the intervention conducted by occupational medicine was always positive, with two different evaluations: for the majority of women (all those who returned to work, and one of those who were unable to return because she had new cancer), the intervention was positive because it provided indications on how to behave in order to manage the reduction of work ability, what the rights were and what the limits of legal protections were and also how to deal with the practical problems that often arise for cancer patients. Furthermore, on the one hand, the consultancy was useful because it indicated to the company the possibility of introducing gradual mechanisms in the return to work, or it made it possible to soften some regulatory indications, such as the exclusion from working at night or the possibility to work at home, which, if not clinically necessary, may be an obstacle to the continuation of one's working career.

Discussion

You have provided an extensive discussion of the findings related to difficulties experienced by women with BCa returning to work. However, this section lacks discussion of the intervention and how the findings of the quantitative data compare to the data from “sample of women of the same age without BC, employed in companies monitored by the occupational doctors who were performing this study” (lines 70-72).

R.: The reviewer is right. In fact, this part of the study was the subject of the first work. In this article, we have briefly indicated some of the results collected at the baseline.

The discussion needs consolidation. It currently reads like a literature review of women with BCa returning to work, rather than a discussion of the finding

R.: We thank the reviewer for this clarification. We have modified the Discussion, prefixing a paragraph that explains the results of the baseline and those of this follow-up study. Furthermore, we have added a few short sentences to explain the logical process we followed. We have divided the Discussion into paragraphs, so that it is easier to recognize the various parts and the sequence of reasoning. Following the indications received, the Discussion is now much clearer and more readable.